# Parental investment and immune dynamics in sex-role reversed pipefishes

**Isabel S. Keller**⊙*, Olivia Roth

GEOMAR, Helmholtz Center for Ocean Research Kiel, Evolutionary Ecology of Marine Fishes, Kiel, Germany

* itanger@geomar.de

**Data Availability Statement:** All relevant data area available from PANGEA database (doi.pangaea.de/10.1594/PANGAEA.921055).

**Funding:** This study was funded by grants to OR from the German Research Foundation (grant: RO 4628/1-1) and the Volkswagen Foundation. The

## Abstract

Parental care elevates reproductive success by allocating resources into the upbringing of the offspring. However, it also imposes strong costs for the care-giving parent and can foster sexual dimorphism. Trade-offs between the reproductive system and the immune system may result in differential immunological capacities between the care-providing and the non-care-providing parent. Usually, providing care is restricted to the female sex making it impossible to study a sex-independent influence of parental investment on sexual immune dimorphism. The decoupling of sex-dependent parental investment and their influences on the parental immunological capacity, however, is possible in syngnathids, which evolved the unique male pregnancy on a gradient ranging from a simple carrying of eggs on the trunk (*Nerophinae*, low paternal investment) to full internal pregnancy (*Syngnathus*, high paternal investment). In this study, we compared candidate gene expression between females and males of different gravity stages in three species of syngnathids (*Syngnathus typhle*, *Syngnathus rostellatus* and *Nerophis ophidion*) with different male pregnancy intensities to determine how parental investment influences sexual immune dimorphism. While our data failed to detect sexual immune dimorphism in the subset of candidate genes assessed, we show a parental care specific resource-allocation trade-off between investment into pregnancy and immune defense when parental care is provided.

## Introduction

Sex-specific life histories have evolved as a consequence of anisogamy; females contribute the large costly eggs and males provide the always available small sperm to reproduction. To gain access to females and to maximize reproductive success, males need to allocate their resources into male-male competition and the display of secondary sexual signals. In contrast, females are striving for longevity to maximize their lifetime reproductive success, which requires a higher investment into pathogen defense [1–5]. Females were thus suggested to having a more efficient immune system than males [6].

Distinct life-history strategies between males and females can also result in differing investment into the upbringing of the offspring, i.e. parental care. Parental care elevates reproductive success by increasing offspring survival, however, it also reduces prospects of future reproduction of the care-providing parent due to a trade-off between investment in offspring and

funders hat no role in study design, data collection and analysis, decision to publish, or preparation of the manuscript.

**Competing interests:** The authors have declared that no competing interests exists.

investment into other fitness-related traits [7]. This can foster sexual dimorphism [8–12] and might also contribute to sexual immune dimorphism [13, 14]. To understand how parental care impacts sexual immune dimorphism, the comparison of closely related species that differ in their provisioning of parental care is essential.

Syngnathids, the teleost family of seahorses, pipefishes and seadragons, with their unique male pregnancy, represent an ideal model system to assess the relationship between parental care and sexual immune dimorphism. Here, closely related species display different forms of paternal care ranging from a rather simple attachment of eggs to the male belly to highly specialized placenta-like structures that provide the embryos with nutrients, oxygen and immunological components [15–20]. In contrast to most species with conventional sex roles, the unique male pregnancy evolution in syngnathids, gives us the opportunity to disentangle sex (i.e., the provisioning of eggs or sperm) from parental care (i.e., pregnancy). This permits to identify if sex, parental care or a combination of both are the drivers for sexual immune dimorphism. To do so, we decoupled the investment in the ability to provide parental care and the investment in provisioning of parental care. The first, investment in the ability to provide parental care, is associated to the evolution of specialized tissues and behaviors. It is solely expressed in the care-providing parent and can be assessed by comparing the immune competence of the care-providing sex and the non-care-providing sex during non-caring phases. The second, investment in provisioning of parental care is associated with the direct provisioning of care and can be measured by comparing caring individuals (i.e., pregnant) with non-caring (i.e., non-pregnant) individuals of the care providing sex.

In this study, we assessed the activity of the immune system in males and females from three species of syngnathids focusing on expression of 29 immune system-related genes and four metabolism-related genes (1). We then compared the sexual immune dimorphism among those closely related species differing in their extent of paternal care provided (2), and last investigated the consequences of provisioning of parental care (i.e., pregnancy) on a subset of immune and metabolism-related genes for each species (3).

For our study, we have chosen three syngnathid species with different extent of paternal care. *Nerophis ophidion* (straight-nosed pipefish) evolved towards a comparably low paternal investment, as females attach their eggs on the abdomen of the male without the development of a placenta-like structurein the male [21]. In the two other species assessed, *Syngnathus rostellatus* (Nilson's pipefish) and *Syngnathus typhle* (broad-nosed pipefish), males brood the eggs in an inverted brood pouch on their tails, and the embryos are provided with nutrients and oxygen over a placenta-like structure. We sampled three male gravity stages (non-pregnant males, pregnant males and males after parturition) and females from all three syngnathid species to measure gene expression of candidate genes with functions in the immune system and related to metabolism to see the implicated resource-allocation trade off.

**(1)** We first examined how the intensity of parental care and the implicated resource allocation trade-off influenced sexual immune dimorphism by comparing non-pregnant males and females of each pipefish species. We proposed that sexual immune dimorphism is positively correlated with the species-specific difference in parental investment between males and females. We hypothesized that in sex-role reversed species females rather allocate their resources into secondary sexual signals than into the immune defense. In contrast, males were hypothesized to allocate their resources into metabolism and the immune system. We thus expected the expression of immune system-related genes and metabolism genes to show a sex-specific pattern.

**(2)** Second, we investigated how the ability to provide parental care influences sexual immune dimorphism. This ability is related to the evolutionary adaptations of life-history strategy displayed in changes of body physiology, behavior and longevity. It can be assessed

when comparing non-pregnant life stages within the care-providing sex to the non-care-providing sex and is hypothesized to rise with the extent of parental investment given. Sex-specific parental investment asks for substantial shifts in life history and resource allocation, implying that fewer resources will be available for other life-history traits. This trade-off can be evaluated by comparing the degree of sexual immune dimorphism among the three sex-role reversed pipefish species that evolved along the paternal investment gradient. We hypothesized that sexual immune dimorphism is linked to the ability to provide parental investment and thus rises with the extent of the evolutionary adaptations to provide parental care. Accordingly, we expected the species without massive evolutionary adaptations, *Nerophis ophidion*, to show the smallest difference in expression of immune system-related and metabolism-related genes between males and females. Stronger differences in gene expression and thus a more distinct sexual immune dimorphism was expected in both *Syngnathus rostellatus* and *Syngnathus typhle* due to substantial adaptations for parental care in body anatomy and neo-functionalization of genes for brood pouch and placenta-like structure development.

(3) Last, we compared the consequences of providing parental care on the care-providing sex. These consequences can only be analyzed during a parental care event (i.e., pregnancy/ brooding). When parental care is provided, resources are needed to develop and sustain the specific tissues, offering a good environment for the offspring and delivering energy towards the offspring [22–24]. During male pregnancy, these resources are not available for other life-history traits. We hypothesized that provisioning costs of parental care (i.e. male pregnancy) induce a resource allocation trade-off in pregnant males and males after parturition. This trade-off is displayed in expression shifts of immune system-related and metabolism-related genes. In terms of immune defense, this entails that non-pregnant males were expected to have distinct gene expression patterns in the immune system-related gene groups as more resources can be allocated towards the immune system compared to both males during pregnancy and males after parturition. Genes related to metabolism were expected to be distinctly expressed during pregnancy as the metabolic rate is highest in this gravity stage. Males after parturition were hypothesized to show a pattern rather similar to pre-pregnant males, as the metabolic rate should return to a pre-pregnant rate in this gravity stage. The above-mentioned pattern was expected to be strongest in *S. typhle*, intermediate in *S. rostellatus* and lowest in *N. ophidion*.

Using syngnathids with their unique male pregnancy, we aimed to validate standing theories of sexual dimorphism and resource allocation trade-offs originally developed for species with conventional sex roles. The comparison of closely-related species with different intensities of paternal care provides novel insights into the transition of parental care in sex-role reversed species and permitted to test the hypothesis that the relationship between paternal investment and sexual dimorphism is not purely dominated by the provisioning of gametes.

## Methods

### Fish sampling

We sampled the three pipefish species, *Nerophis ophidion*, *Syngnathus rostellatus* and *Syngnathus typhle* in April and May 2016 by snorkeling in seagrass meadows around the Kiel Fjord (54˚44'N; 9˚53'E). Upon catching, animals were transferred to our aquaria facilities at GEOMAR, kept at 18˚C and 15 PSU (Baltic Sea summer conditions, 14/10 light/dark cycle) in a circulation system, and fed twice a day with live and frozen artemia, mysids and copepods. All animals were kept for acclimatization to laboratory conditions for two weeks in the aquaria at GEOMAR. Males were randomly assigned to one of the three gravity stages: non-pregnant males, pregnant males and males after parturition. Only males assigned to be sampled at

pregnant and after parturition stage were permitted to mate with the females. At mid pregnancy, when Syngnathus embryos had hatched from the egg within the brood pouch, males were either left in the tanks to be sampled after parturition or sampled immediately. All animals were handled according to the animal welfare laws of Germany, under a permit from the "Ministerium für Landwirtschaft, Umwelt und ländliche Räume des Landes Schleswig-Holstein" called „Sexueller Immundimorphismus in Seenadeln und Buntbarschen entlang eines Brutpflege Gradienten (V242-7224.121–19 (141-10/13))". Fish were killed by an overdose of Tricaine mesylate (MS-222, Syndel) and their weight and length were measured. Gills were collected and stored in RNAlater (Qiagen) for candidate gene expression analysis.

## Gene expression assays

RNA was extracted from gill tissue with RNeasy 96 Universal Tissue Kit (Qiagen) following the manufacturers protocol for animal tissues. RNA yield was measured by spectrometry (NanoDrop ND-1000; peQLab) and 200 ng/µl were used for reverse transcription with QuantiTect®Reverse-Transcription Kit (Qiagen). We used primer pairs for immune system-related and metabolism-related genes [16] (list of all primers see S1 Table). The expression patterns of genes were measured using a Fluidigm-BioMarkTM system based on 96.96 dynamic arrays (GE-Chip) following the protocol attached in S1 Protocol.

## Data analysis & statistics

Gene expression data were processed using the Fluidigm-integrated software (Fluidigm Real-Time PCR analysis; BioMark Version 4.1.2). Samples with melt curves deviating in mean temperature from the mean melt curve per gene were excluded. Resulting in 83 samples (8 *N. ophidion* non-pregnant males, 7 *N. ophidion* pregnant males, 5 *N. ophidion* males after parturition and 6 *N. ophidion* females; 8 *S. rostellatus* non-pregnant males, 7 *S. rostellatus* pregnant males, 8 *S. rostellatus* males after parturition and 8 *S. rostellatus* females; 6 *S. typhle* non-pregnant males, 5 *S. typhle* pregnant males, 8 *S. typhle* males after parturition and 7 *S. typhle* females) for the gene expression analysis. Mean cycle threshold (Ct), standard deviation (SD), and coefficient of variance (CV) were calculated for each remaining sample duplicate. Samples with a CV lower than 4% were replaced by the mean value over all samples per gene. *Hivep2* and *hivep3* with the lowest geNorm (qbase+ version 3.0, biogazelle) values, indicating the most constant expression over all treatments, were chosen as reference genes. For relative gene expression, the geometric mean of the two reference genes (*hivep2* and *hivep3*) was subtracted from the negative mean Ct value of interest per sample resulting in– ΔCt values.

All statistical analyses were done in R version 3.1.3. GUI 1.65. Statistical analysis of gene expression was done by calculating a PERMANCOVA ((*adonis (x~ y+K, method = "euclidean", permutations = 1000)*) with the factors "sex"or "gravity stage", respectively and the condition factor (K, weight / length$^3$) as covariable. For each species (*N.ophidion*, *S. rostellatus* and *S.typhle*) and each factor ("sex" or "gravity stage") all gene groups (adaptive immune genes, innate immune genes, complement component genes, metabolism-related genes) were tested independently via the above mentioned PERMANCOVA. Using a principal component analysis (PCA, FactoMineR, [25]) gene contribution to the principal component 1 and principal component 2 (PC1 and PC2) have been calculated (((contribution [%] PC1*eigenvalue PC1) + (contribution [%] PC2*eigenvalue PC2))/ (eigenvalue PC1 + eigenvalue PC2)). Genes with a combined contribution to the variance explained by PC1 and PC2 higher than expected by chance (100/ (# of genes)) were included in the visualization.

**Table 1. Species-specific PERMANCOVA results of candidate gene expression assessing sex differences.** A condition factor ($K = W/TL^3$) was included as covariable.

| *Nerophis ophidion* | | | | | |
| --- | --- | --- | --- | --- | --- |
| Gene categories | Model | Sex | | Condition Factor | |
| | R2 | F. Model | Pr (>F) | F. Model | Pr (>F) |
| Adaptive IS genes | 0.88 | 0.97 | 0.425 | 0.53 | 0.766 |
| Innate IS genes | 0.87 | 0.98 | 0.400 | 0.73 | 0.565 |
| Complement components genes | 0.79 | 1.03 | 0.357 | 1.91 | 0.175 |
| Metabolism related genes | 0.85 | 0.82 | 0.446 | 1.09 | 0.349 |
| Df Residuals / Model | 11 | 1 | | 1 | |
| Df Total | 13 | | | | |
| *Syngnathus rostellatus* | | | | | |
| Gene categories | Model | Sex | | Condition Factor | |
| | R2 | F. Model | Pr (>F) | F. Model | Pr (>F) |
| Adaptive IS genes | 0.88 | 0.77 | 0.515 | 0.96 | 0.396 |
| Innate IS genes | 0.85 | 0.66 | 0.657 | 1.62 | 0.146 |
| Complement components genes | 0.82 | 1.33 | 0.257 | 1.59 | 0.207 |
| Metabolism related genes | 0.94 | 0.53 | 0.662 | 0.29 | 0.780 |
| Df Residuals / Model | 13 | 1 | | 1 | |
| Df Total | 15 | | | | |
| *Syngnathus typhle* | | | | | |
| Gene categories | Model | Sex | | Condition Factor | |
| | R2 | F. Model | Pr (>F) | F. Model | Pr (>F) |
| Adaptive IS genes | 0.86 | 1.07 | 0.442 | 0.62 | 0.756 |
| Innate IS genes | 0.74 | 1.18 | 0.281 | 2.35 | 0.045 |
| Complement components genes | 0.70 | 2.87 | 0.061 | 1.39 | 0.270 |
| Metabolism related genes | 0.80 | 1.62 | 0.204 | 0.95 | 0.463 |
| Df Residuals / Model | 10 | 2 | | 1 | |
| Df Total | 12 | | | | |

## Results

### (1) Sexual immune dimorphism in species with different parental care intensities

Multivariate analysis (PERMANCOVA) did not show effects of sex (male vs. female) on the expression of any of the four gene groups (adaptive immune system genes, innate immune system genes, complement component genes and metabolism-related genes) regardless of the species assessed (Table 1). As such, neither *Nerophis ophidion*, *Syngnathus rostellatus* nor *Syngnathus typhle* show sexual dimorphism in gene expression patterns of the measured immune system-related or metabolism-related genes.

### (2) Influence of adaptations for parental care on the immune system

Influences of adaptations to parental care could be seen by quantifying the delta in gene expression between the sex providing parental care and the sex not providing parental care within one species. This delta was then compared across species. The species with more intense parental care was expected to have more pronounced life-history adaptations towards the ability to provide parental care in the care-providing sex and was thus suggested to have a stronger sexual dimorphism. As we could not identify differences in gene expression pattern between males (the sex providing parental care) and females (the sex not providing parental care), we

**Table 2. Species-specific PERMANCOVA results of candidate gene expression assessing differences in male gravity stages.** A condition factor (K = W/TL$^3$) was included as covariable. Significant results are written in bold letters.

| **A** *Nerophis ophidion* | | | | | |
|---|---|---|---|---|---|
| Gene categories | Model | Gravity stage | | Condition Factor | |
| | R2 | F. Model | Pr (>F) | F. Model | Pr (>F) |
| Adaptive IS genes | 0.79 | 1.78 | 0.098 | 0.03 | 0.569 |
| Innate IS genes | 0.77 | 1.80 | 0.076 | 1.12 | 0.369 |
| Complement components genes | 0.74 | 1.18 | 0.349 | 3.30 | 0.055 |
| Metabolism related genes | 0.61 | 4.42 | **0.003** | 1.22 | 0.325 |
| Df Residuals / Model | 16 | 2 | | 1 | |
| Df Total | 19 | | | | |
| **B** *Syngnathus rostellatus* | | | | | |
| Gene categories | Model | Gravity stage | | Condition Factor | |
| | R2 | F. Model | Pr (>F) | F. Model | Pr (>F) |
| Adaptive IS genes | 0.62 | 4.09 | **0.002** | 3.44 | **0.017** |
| Innate IS genes | 0.81 | 1.49 | 0.120 | 1.62 | 0.158 |
| Complement components genes | 0.86 | 0.80 | 0.571 | 1.49 | 0.203 |
| Metabolism related genes | 0.93 | 0.61 | 0.699 | 0.15 | 0.916 |
| Df Residuals / Model | 19 | 2 | | 1 | |
| Df Total | 22 | | | | |
| **C** *Syngnathus typhle* | | | | | |
| Gene categories | Model | Gravity stage | | Condition Factor | |
| | R2 | F. Model | Pr (>F) | F. Model | Pr (>F) |
| Adaptive IS genes | 0.76 | 2.19 | **0.027** | 0.31 | 0.942 |
| Innate IS genes | 0.73 | 2.50 | **0.037** | 0.55 | 0.694 |
| Complement components genes | 0.63 | 4.20 | **0.005** | 0.37 | 0.763 |
| Metabolism related genes | 0.81 | 1.57 | 0.196 | 0.28 | 0.803 |
| Df Residuals / Model | 15 | 2 | | 1 | |
| Df Total | 18 | | | | |

could not quantify the influence of life-history adaptations for the ability to provide parental care.

## (3) Influence of parental care provisioning on the immune system

Multivariate analyses (PERMANCOVA) showed effects of male gravity stages on the expression of genes from several functional groups differing among species according to extent of male pregnancy. Principal component analysis showed differential gene expression patterns according to male gravity stage and visualized the contribution of the measured genes in each of the three species.

In *Nerophis ophidion*, the species with the lowest investment to male pregnancy, multivariate analyses (PERMANCOVA) revealed differential gene expression depending on male gravity stage in the gene group metabolism-related genes (Table 2A).

In the principal component analysis of the metabolism-related genes (Fig 1) the first two PC axes explain 82.7% of the total variance (PC1 explains 58.4%, PC2 explains 24.2%). The individual grouping according to male gravity stage showed a differential gene expression between non-pregnant males and both pregnant males and males after parturition along the first principal component (PC1 58.4% of the total variance, clouds are 95% confidence ellipses). In addition, here, males after parturition and pregnant males show no difference in

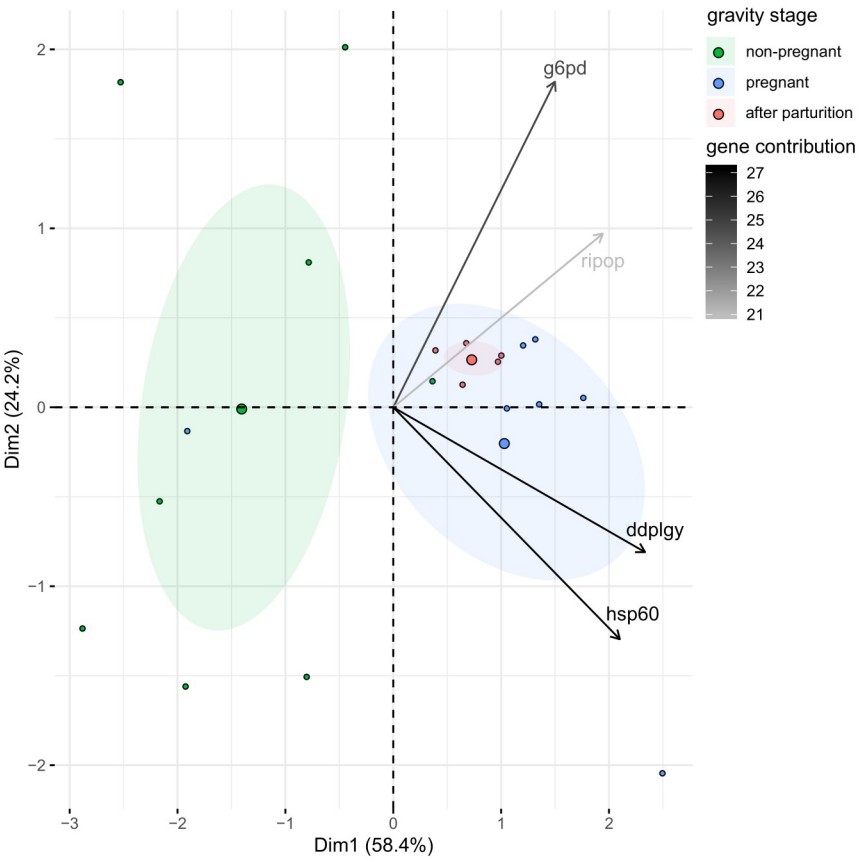

**Fig 1. Principal component analysis (PCA) of effects of male gravity stages on metabolism-related gene expression in *N.ophidion*.** The first two principal components, PC1 explaining 58.4% of the total variance and PC2 explaining 24.2% of the total variance are shown together with 95% confidence ellipses (shaded area) around a center of gravity (big points). Samples are indicated as small points. Colors represent male gravity stages; green for non-pregnant males, blue for pregnant males and red for males after parturition. Shading of the arrows show contributions of the depicted genes to the principal components.

metabolism-related gene expression. Within the metabolism-related gene group, two genes of four have a higher contribution [%] to those two principal components than expected by chance (expected contribution [%] = 100/4 = 25%) (S2 Table).

In the species with intermediate parental investment in male pregnancy, *Syngnathus rostellatus*, the multivariate analysis (PERMANCOVA) showed differential expression of adaptive immune system genes (Table 2B).

In the principal component analysis of genes associated with the adaptive immune system (adaptive immune system genes) the first two principal components describe 56.6% of the total variance split to principal component 1 (PC1) explaining 33.4% of the total variance and Principal component 2 (PC2) explaining 23.2% of the total variance (Fig 2). An individual grouping according to male gravity stages reveals that both non-pregnant males and males after parturition share a similar gene expression pattern, whereas pregnant *S. rostellatus* differ in their gene expression pattern from the latter two. Five genes have a higher contribution [%] than expected (100/11 = 9.095%) see S2 Table.

In *Syngnathus typhle*, the species with the highest investment in paternal pregnancy, the multivariate statistics (PERMANCOVA) showed differences in gene expression patterns according to gravity stage in all gene groups assessed except in metabolism-related genes

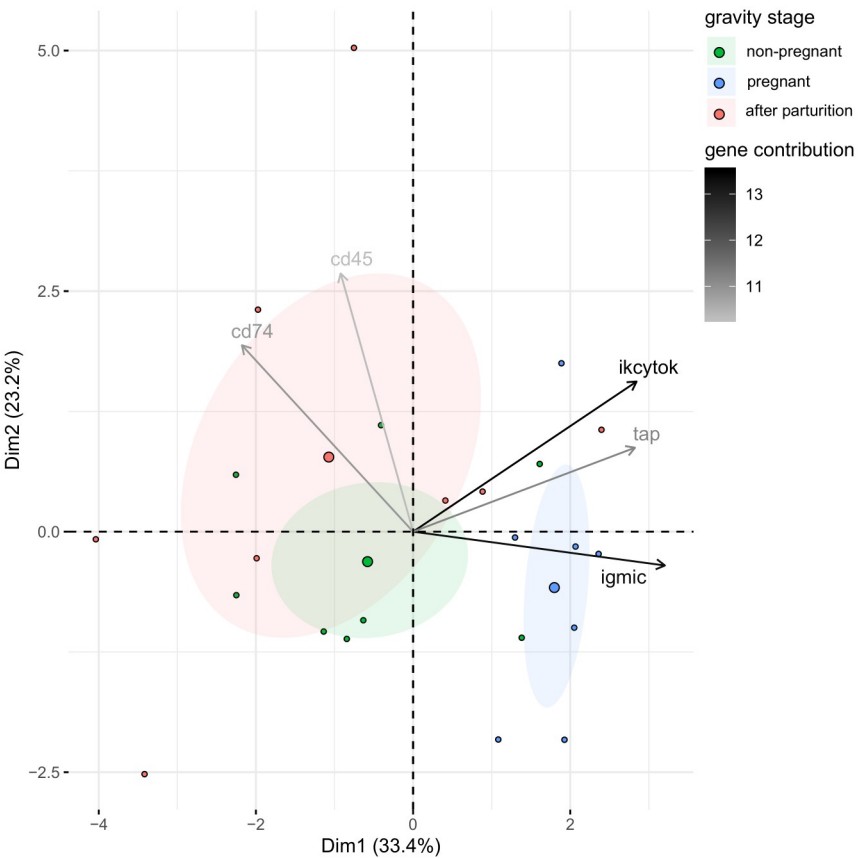

**Fig 2. Principal component analysis (PCA) of effects of male gravity stage on adaptive immune system gene expression in *S. rostelltaus*.** The first two principal components, PC1 explaining 33.4% of the total variance and PC2 explaining 23.2% of the total variance, are shown with 95% confidence ellipses (shaded area) around a center of gravity (big points). Samples are indicated as small points. Colors represent male gravity stages; green for non-pregnant males, blue for pregnant males and red for males after parturition. Shading of the arrows show contributions of the genes with a contribution to the principal components higher than expected.

(Table 1C). The principal component analyses in groups with differential expression show two main patterns of gene expression, a distinct expression of pregnant males and males after parturition but both not differing from the expression pattern of non-pregnant males found in the groups innate immune system genes and complement component genes. In addition, a distinct expression of pregnant males compared to the other two gravity stages in the gene group adaptive immune system genes.

Principal component analysis of the gene expression pattern from genes associated with the innate immune system explains a combined variation of the first two principal components of 54.5% split on the principal component 1 (PC1, 37.6%) and principal component 2 (PC2, 16.8%) (Fig 3A). The overall gene expression pattern shows differential expression of males after pregnancy from males during pregnancy along the PC1, whereas non-pregnant males did not differ in gene expression compared to both other groups. Six genes have a higher contribution than expected (EC = 100/15 = 6.7%) (S2 Table).

Principal component analysis of the genes from the complement component genes group show a similar gene expression pattern in the individual grouping according to male gravity stage as the genes associated to the innate immune system (innate immune system genes). The

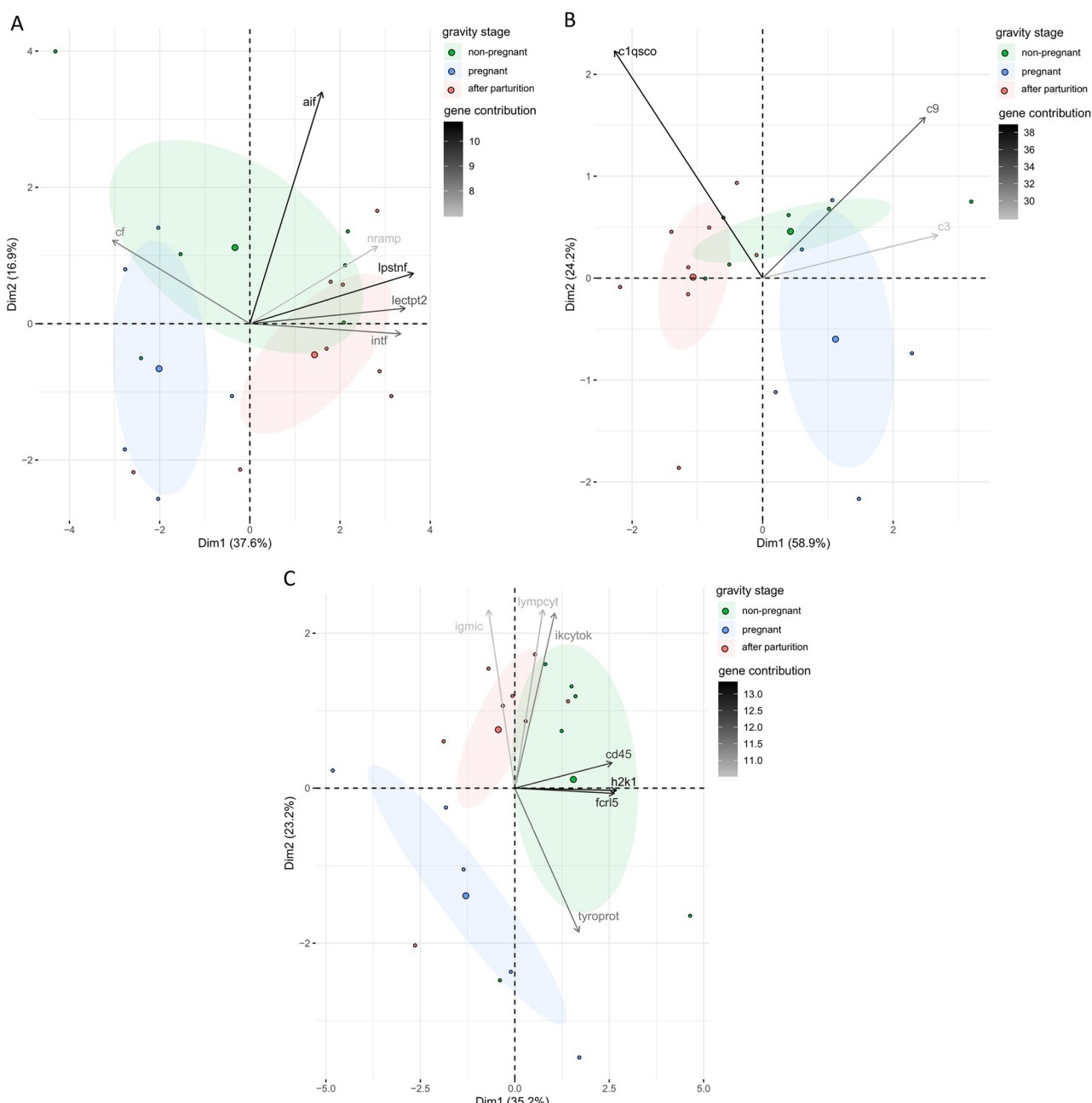

**Fig 3. Principal component analysis (PCA) of effects of male gravity stage on gene expression in *S. typhle*.** The first two principal components, PC1 explaining 37.6% of the total variance and PC2 explaining 16.8% of the total variance, are shown with 95% confidence ellipses (shaded area) around a center of gravity (big points). Samples are indicated as small points. Colors represent male gravity stages; green for non-pregnant males, blue for pregnant males and red for males after parturition. (**A**) PCA of effects of male gravity stage on innate immune system genes. Shading of the arrows shows contributions of the genes with a contribution to the principal components higher than expected. (**B**) PCA of effects of male gravity stage on complement component genes. Shading of the arrows shows contributions of the depicted genes to the principal components. (**C**) PCA of effect of male gravity stage on adaptive immune system genes. Shading of the arrows shows contributions of the genes with a contribution to the principal components higher than expected.

first two principal components explain 58.9% (PC1) and 24.2% (PC2) of the variance in the system and explain a combined variance of 83.2%. (Fig 3B).

A gene expression pattern differential to the pattern seen in the innate immune system genes and the complement component genes is visible in the principal component analysis of adaptive immune system-related genes (adaptive immune system genes). The total variance explained by the first two principal components is 58.3% whereas the principal component 1 (PC1) explains 35.2% of the variation and the principal component 2 (PC2) explains 23.2% of the total variation (Fig 3C). The individual grouping according to male gravity stages shows differential gene expression patterns of pregnant males and both males after parturition and non-pregnant males. Males after parturition and non-pregnant males, however, do not differ in their gene expression patterns. Of the 11 genes in the gene group adaptive immune system genes seven show a higher contribution [%] than expected by chance (100/11 = 9.09%) (S2 Table).

Genes with a higher contribution to the respective expression patterns than expected by chance and expressed in a similar way in both *S. rostellatus* and *S. typhle*, the two species with intensive paternal pregnancy, could give a closer insight into the direction of gene expression during male pregnancy and highlight the most important genes also for future research. Such genes can be found in the adaptive immune system gene group as this group shows differential expression depending on male gravity stages in both *S. rostellatus* and *S. typhle*. Three genes, *cd45*, *igmlc* and *ikcytok*, have a high contribution to the gene expression patterns in both *S. rostellatus* and *S. typhle*. In *S. rostellatus* (Fig 4A), *cd45* has a lower relative expression in pregnant stages compared to the other gravity stages, whereas both other genes of interest, *igmic* and *ikcytok*, show an elevation of the relative expression during pregnancy compared to the other male gravity stages. In *S. typhle* (Fig 4B) the expression pattern of *cd45* is similar as in *S. rostellatus*, thus, the relative expression is lower in pregnant males compared to the other pregnancy stages. In contrast to *S. rostellatus*, however, the relative expression of *igmic* and *ikcytok* is lower in pregnant males of *S. typhle* compared to the other gravity stages.

## Discussion

We addressed the relationship between sexual immune dimorphism and paternal investment along a gradient of paternal care intensity in three species of sex-role reversed pipefishes. We asked three questions regarding the influence of parental investment intensity on the expression patterns of immune system-related and metabolism-related genes in these species. First, does parental care affect sexual immune dimorphism? Second, do adaptations necessary to provide parental care influence the immune system? Third, how does the immediate resource-allocation trade off during a phase of intense parental care influence the immune system of the care-providing sex? To this end, we measured candidate gene expression in three species of sex-role reversed pipefishes with different extent of paternal investment ranging from low paternal investment (*Nerophis ophidion*) over intermediate parental investment (*Syngnathus rostellatus*) to high parental investment (*Syngnathus typhle*).

We first hypothesized that in a sex-role reversed system, non-pregnant males would have a higher immunological activity than females, as the males enhance their investment into the immune system to ensure longevity [13]. We could, however, not identify sexual dimorphism in the expression of immune system-related genes or metabolism-related genes in any of the three species measured. This is astonishing, as sexual immune dimorphism has been shown in earlier studies with *Syngnathus typhle* and in the phylogenetically closely-related seahorse species *H. erectus* [13, 26]. Both studies tested for sex differences in immune parameters by measuring the amount of certain immunologically active components (Immunoglobulin M (IgM),

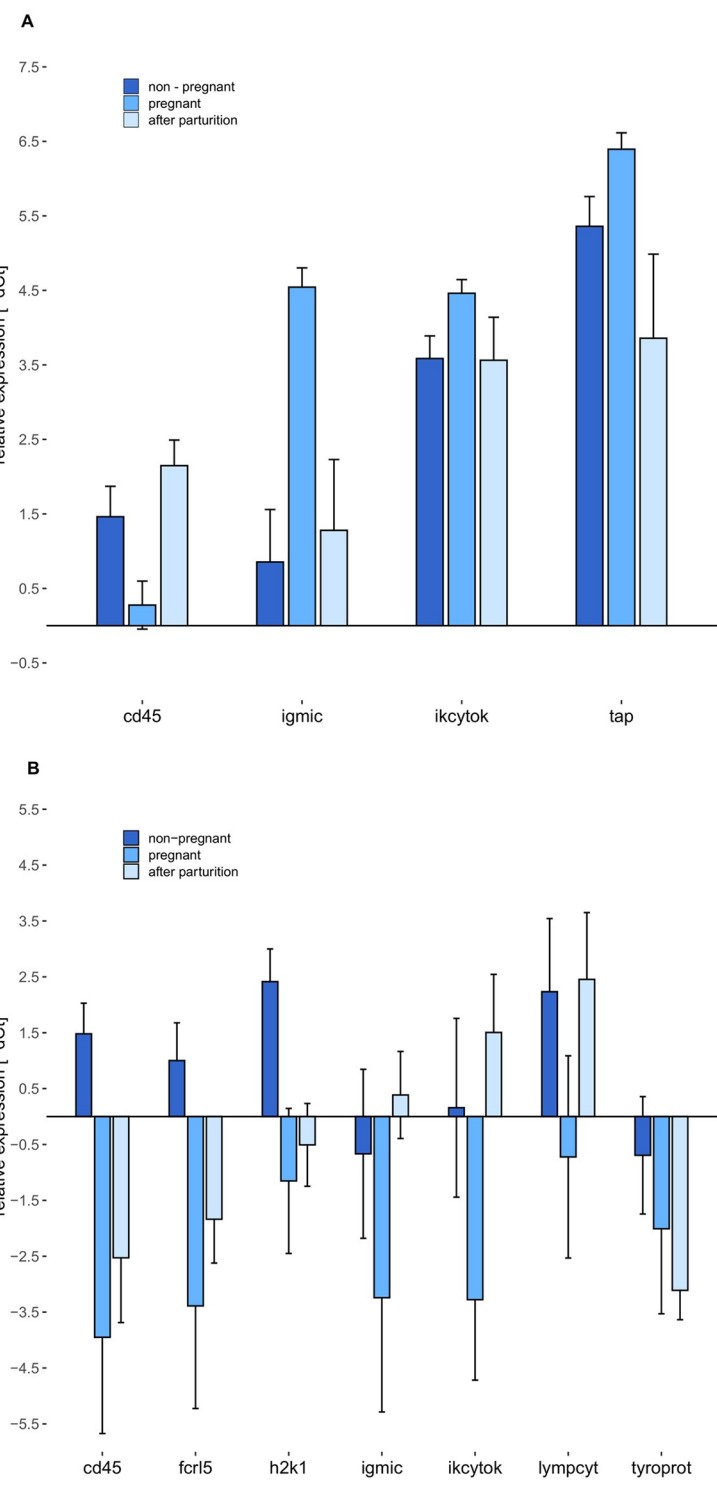

**Fig 4. Effects of male gravity stages on gene expression of relevant adaptive immune system genes.** Both graphs show the group mean and standard error of the relative gene expression (-dCt) of adaptive immune system genes with a higher contribution to the respective PCA than expected by chance. Non-pregnant males are shown in dark blue, pregnant males in blue and males after parturition in light blue. (**A**) Effects of male gravity stages on gene expression in *S. rostellatus*. (**B**) Effects of male gravity stages on gene expression in *S. typhle*.

interferon-alpha (IFN-alpha), interleukin 2 (IL-2), lysozyme (LZM), or respiratory burst and antimicrobial activity measurements) and by measuring cells associated to the immune system (proportion of monocytes / lymphocytes, proliferation of lymphocytes and phagocytic capacity of leucocytes). Both studies measured cellular or humoral immune parameters and not the underlying gene expression patterns. In contrast, we here measured the expression of genes associated to the immune system, which may explain the distinct results. Gene expression patterns cannot be directly translated to the activity of immunological effectors / parameters, as many genes may be involved in the pathway towards an activity change of an immunological parameter. Further, not all gene expression changes recorded in an individual at a certain time point may actually translate into functional proteins. Additionally, for gene expression studies we could only choose a small subset of possible genes due to feasibility reasons. Here the subset of 33 genes, chosen due to their important and known functions in the immune system, might not include those genes associated with sexual immune dimorphism. These immune genes might not be under strong sex-specific selection and might thus be expressed similarly in both sexes. For a better insight into the genes underlying sexual immune dimorphism, a whole transcriptome study is the next step to undertake.

Second, we hypothesized that the ability to provide intense parental care, measured as difference in sexual dimorphism between species, increases with the extent of adaptation to paternal care. As we were not able to detect sexual immune dimorphism in gene expression, we cannot conclude that adaptations to intense parental care have lasting influences on the immune system. However, this theory of evolutionary adaptations to parental care and their influence on the parental immune system is strengthened by a novel hypothesis in mammalian sexual immune dimorphism research. The pregnancy compensation hypothesis states that evolutionary adaptations to the toleration of an embryo as a non-self-tissue within the female body during pregnancy leads to differential expression and actions of the female immune system compared to the male immune system actions [27]. This novel finding shows, that the theoretical background of this part of the study has merit in a human model system and that further testing of this hypothesis with alternative methods might give a similar insight into other species ultimately enhancing the understanding of the evolution of sexual immune dimorphism.

Third, we hypothesized that males pay a provisioning cost during pregnancy displayed in a differential expression pattern between non-pregnant males, pregnant males and males after parturition depending on the intensity of the parental care provided (decreasing effects from *S.typhle* over *S. rostellatus* to *N. ophidion*). Our results indeed suggest that expression changes of immune system-related genes and metabolism-related genes depend on the intensity of paternal care. In *N. ophidion*, the species with the lowest investment in paternal care, the group of metabolism-related genes is affected in its expression by the provisioning of parental care, being differentially expressed in non-pregnant males compared to both pregnant males and males after parturition. Interestingly, metabolism-related genes do not seem to be affected in their expression in either of the other two species, *S. typhle* or *S. rostellatus*, this could suggest a metabolic cost of paternal care in *N. ophidion* that is stronger than the metabolic cost in *S. typhle* or *S. rostellatus* despite their more intense pregnancy. This is supported by previously documented metabolic costs (high paternal weight loss during care for small weight brood) of parental care in *Nerophis lumbriciformis* [21]. It is tempting to speculate about a potential relationship between the adaptation to provide parental care and the metabolic demand issued by parental care. As such, species with less-specialised adaptation of life-history traits to parental care could eventually have a higher cost associated with the provisioning of parental care.

*Syngnathus rostellatus* shows a distinct expression of adaptive immune system genes but not of metabolism-related genes. This hints towards a higher investment in terms of the

immune system in *S. rostellatus* compared to *N. ophidion*, which one may interpret as depiction of extensive differences in parental care intensities such as internalisation of paternal care and specialized embryonal provisioning.

The gene expression pattern is affected by male gravity stage in *S. rostellatus* illustrated by a distinct gene expression in pregnant *S. rostellatus* compared to non-pregnant males and males after parturition. This, however, also suggests a fast recovery to a state similar to pre-pregnancy possibly attributed to the close contact with the juveniles. A modulation of the immune system might be necessary to prevent juvenile rejection but also to allow transfer of immune components from the male to its offspring. One of the three genes with high contribution in both syngnathid species, *cd45*, is associated with fetal rejection and subsequent miscarriage in human pregnancy upon upregulation [28]. Thus, a downregulation of *cd45* expression could be important in syngnathids for the tolerance of the embryos and the completion of a successful male pregnancy.

In *Syngnathus typhle*, we see effects of male gravity stage on the expression of all gene groups except the metabolism-related genes group. In the gene groups affected by male gravity stage, two distinct gene expression patterns can be found in *S. typhle*. The first pattern includes the innate immune system genes and the complement component genes; their expression is affected by male gravity stage in *S. typhle*, a pattern that cannot be revealed in the other two species assessed (*N. ophidion* and *S. rostellatus*). In those gene groups pregnant males and males after parturition do differ in their gene expression pattern, both are, however, not different from non-pregnant males. As in *N. ophidion* this could hint towards a depletion of resources during pregnancy that hampers with a fast recovery of the male after pregnancy has ceased. In contrast to *N. ophidion*, and possibly attributed to the more intense male pregnancy, this effect is visible in components of the immune system. The second pattern, visible in the gene expression of the adaptive immune system genes, shows differences in gene expression between pregnant males and both males after parturition and non-pregnant males. This pattern is similar to what we found in *S. rostellatus*. A differential expression of adaptive immune system genes could have arisen in order to be able to protect juveniles from pathogens and transfer immune components to the juveniles. Further, this pattern could be attributed to the downregulation of the adaptive immune system to establish the close internal connection to the juveniles necessary to prevent juvenile tissue rejection. This differential expression of adaptive immune system genes during intense male pregnancy could hint towards evolutionary adaptations to paternal pregnancy. Interestingly, genetic modification suggested to be in relation with the acceptance of juvenile tissues during internal pregnancy has been shown in *Syngnathus* with the loss of MHCII and associated pathways important for self- non-self-recognition [29, 30].

Our data show that sexual immune dimorphism in species with sex-role reversal cannot be detected by the set of genes used in this study. Thus, a prediction of the influence of adaptations to intense parental care cannot be drawn using this method. Anyway, we can see distinct involvement of the immune system when the different forms of male pregnancy are considered. As such, low intensity of paternal investment does rather affect metabolic pathways whereas with rising male pregnancy intensity stronger effects on immune gene expression patterns can be found. Especially the adaptive immune system genes seem to be affected by an ongoing internal male pregnancy with a placenta-like system. A prevention of embryo rejection as non-self-tissue or potential immune priming of the offspring might explain the identified pattern. Both reasons could be important at different stages of the pregnancy and even occur in combination.

In the future, comparative transcriptomics should shed light on the sexual immune dimorphism reflected in gene expression patterns, which also permits to gain insight on the influence

of sex-specific adaptations to parental care strategies beyond anisogamy. Further, a finer resolution of sampling during pregnancy could highlight reasons (embryo rejection and/or provisioning purposes) of differential adaptive immune-gene expression patterns. Assessing a broader diversity of syngnathid species would permit to dig deeper in the connection of parental investment and sexual immune dimorphism along the male pregnancy gradient. Such a finer resolution of investment in evolutionary adaptations to specialized parental care patterns and investment in provisioning of parental care along the transition of differential paternal pregnancy modes would provide insight in the evolution of sex-role reversal and the corresponding paternal care intensity.

## Supporting information

**S1 Table. List of all primer used for gene expression analysis.** Sorting of primer to the respective gene groups are shown here together with the gene name and the forward and reverse primer sequence. Origin and reference to theses primers are listed in the last column.
(XLSX)

**S2 Table. Gene contribution to the first two principal components of the PCA for effects of male gravity stage on gene expression.** Contribution of each gene is shown within its functional gene group and separately for each species. Names of the gene groups are written as horizontal header including the expected contribution value (EC). Bold headers depict significance in the Multivariate Analysis. Bold gene contribution visualizes a gene contribution higher than the expected contribution by chance.
(XLSX)

**S1 Protocol. Protocol gene expression using a Fluidigm-BioMarkTM system based on 96.96 dynamic arrays (GE-Chip).** Details on Preamplification of target cDNA, sample mix and assay mix and GE-chip loading technique.
(DOCX)

## Acknowledgments

We thank D. Gill for support in the laboratory, A. Beemelmanns for providing the primers used in this study and W. Salzburger for his comments on an earlier version of this manuscript. In addition, we thank F. Wendt for building and maintaining the aquaria facilities.

## Author Contributions

**Data curation:** Isabel S. Keller.

**Formal analysis:** Isabel S. Keller.

**Funding acquisition:** Olivia Roth.

**Investigation:** Isabel S. Keller.

**Methodology:** Isabel S. Keller, Olivia Roth.

**Resources:** Olivia Roth.

**Software:** Isabel S. Keller.

**Supervision:** Olivia Roth.

**Validation:** Olivia Roth.

**Visualization:** Isabel S. Keller.

**Writing – original draft:** Isabel S. Keller.

**Writing – review & editing:** Isabel S. Keller, Olivia Roth.

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
