## [Decision Letter · Decision Letter 0]

1 Jun 2020

PONE-D-20-02201

Parental investment and immune dynamics in sex-role reversed pipefishes

PLOS ONE

Dear Dr. Tanger,

Thank you for submitting your manuscript to PLOS ONE. After careful consideration, we feel that it has merit but does not fully meet PLOS ONE’s publication criteria as it currently stands. Therefore, we invite you to submit a revised version of the manuscript that addresses the points raised during the review process.

Two experts in the field have reviewed the manuscript. Both expressed high enthusiasm for it, but they also have provided comments as to how it can be improved in terms of the scientific statements, grammar, and figures. 

We look forward to receiving your revised manuscript.

Kind regards,

Cheryl S. Rosenfeld, DVM, PhD

Academic Editor

PLOS ONE

Journal Requirements:

Additional Editor Comments (if provided):

Reviewers' comments:

Reviewer's Responses to Questions

**Comments to the Author**

1. Is the manuscript technically sound, and do the data support the conclusions?

Reviewer #1: Yes

Reviewer #2: Yes

2. Has the statistical analysis been performed appropriately and rigorously? 

Reviewer #1: Yes

Reviewer #2: Yes

3. Have the authors made all data underlying the findings in their manuscript fully available?

Reviewer #1: Yes

Reviewer #2: No

4. Is the manuscript presented in an intelligible fashion and written in standard English?

Reviewer #1: Yes

Reviewer #2: Yes

5. Review Comments to the Author

Reviewer #1: I have read your manuscript with great interest, as I find the topic fascinating. The text is clear and easy to follow (at least to me), the methodology is appropriate, the figures are informative and the results/discussion very interesting. As such, I congratulate the authors on a really nice work. I list bellow some questions and very minor suggestions that might, if the authors agree, possible slightly improve the manuscript.

P3 L68: “Syngnathids, the teleost family of seahorses and pipefishes, …”

Unimportant but, to be more wide-ranging, would you consider including at least the seadragons?

P4 L95: “Nerophis ophidion (straight-nosed pipefish) with a comparably low paternal investment, as females simply attach their eggs on the abdomen of the male without any additional provisioning of paternal investment”.

If you are stating that females don’t invest further once the oocytes are passed onto the male, I agree. If you are stating that the males don’t provision during pregnancy, I have my doubts. Either way, maybe you could refine this sentence in order to avoid any hypothetical misperception.

P9 L208: “Statistical analysis of gene expression was done calculating two PERMANCOVA…”

Sounds weird… was done by? Or something like that.

P10 L229: “… nor Syngnathus typhle show a sexual dimorphism in gene expression…”

Maybe remove the ‘a’ before sexual dimorphism…

P17 L368: repeated ‘in the’

P20 L431: “Second, we hypothesized that the adaptation to intense parental care, measured as difference in sexual dimorphism between species, increases with the extent of adaptation to paternal care”.

Don’t really understand this sentence…

P21 L479: “In Syngnathus typhle, we can see effects of male gravity stage on…”

As you probably detected by now, I am not a native English speaker. As such, my ability to provide significant improvements to the text is very slim. Nevertheless, I was wondering if the term ´gravity´ is really a synonym of pregnancy… If it is, I am sorry to have wasted your time.

P21 L460: “As such, species with less-specialised adaptation of life-history traits to parental care could eventually have a higher cost associated with the provisioning of parental care”.

I don’t disagree with this ‘speculation’ (as characterized by the authors), at all. Indeed, this would be something interesting to research further (there is not much information on syngnathid male investment during pregnancy). In a previous study (Pregnant pipefish with a simple brooding surface lose less weight when carrying heavier eggs: evidence of compensation for low oocyte quality?), I found that in a different Nerophis species (N. lumbriciformis), males seem to invest considerably during pregnancy, especially so when eggs are of poorer quality. I’m not asking for a citation (really!), just saying that your ‘speculation’ seems to be a nice hypothesis (backing up the observed change in the expression of metabolism-related genes group in Nerophis).

P22 L486: “As in N. ophidion this could hint towards a depletion of resources during pregnancy that hamper with a fast recovery of the male after pregnancy has ceased.”

I agree. Having worked with both Syngnathus and Nerophis, we see that Nerophis males that much more time to get pregnant again. For instance, Syngnathus abaster males can become pregnant rapidly after giving birth while N. lumbriciformis males need many days (in the lab, more than a week, at best). In the field, based on their migration patterns to and from the intertidal, they seem to take a month or more to become pregnant again.

P6 L150: “The above-mentioned pattern was expected to be strongest in S. typhle, intermediate in S. rostellatus and…”

Out of curiosity, what do you think is the main driver(s) of the differences observed between the two Syngnathus species?

P22 L499: male instead of mal

P7 L171: “At mid pregnancy, …”

Since pipefish (those with a marsupium) pregnancy can, roughly, be sub-divided into two periods (embryos within the egg and outside the egg but still inside the male’s brood pouch), and male investment can differ within these two stages (I imagine that male investment should be more critical once the embryos hatch, having spent most of the mother-derived resources), I would ask if 1) the embryos from both Syngnathus species were within the same stage of development? and 2) the embryos were still within the egg or already free in the pouch? Personally, I think it would be helpful to state this information in the methods. If I’m correct, and male investment is stronger once the embryos hatch (I think I discuss something along these line in the “woman in red” paper), sampling before and after embryo hatching would produce very different results.

P23 L513: “A prevention of embryo rejection as non-self-tissue or potential immune priming of the offspring might explain the identified pattern”.

I agree. This being said, what I’m about to ask adds nothing to your manuscript, but I imagined I have a chance to hear your opinion, in case you are willing to share your thoughts on the subject. Since fertilization occurs within the brood pouch, a sneaker male strategy (never found… probably for a reason) would be nearly impossible (a sneaker male would have a huge task, Tom Cruise ´mission impossible´ style, to be able to inject his few spermatozoa together with a female laying her oocytes. The time window would be extra-small). In Nerophis, although difficult, a sneaker male (I have been unable to find any instances of this as well) would probably have more time and easier access to unfertilized eggs. So, do you think that the immune system somehow mirrors this difference in the chance of rearing a embryo that is from a different father?

Again, compliments on your work.

Nuno Monteiro

Reviewer #2: Does the manuscript adhere to the PLOS Data Policy? Additional details can be found at http://journals.plos.org/plosone/s/materials-and-software-sharing.

I don’t know

This manuscript unites a strong and fascinating conceptual foundation with the unique biological system of syngnathids – which exhibit male pregnancy across a spectrum of investment intensities – to investigate causes of sexual dimorphism in immune function. As the authors astutely note (lines 75-76), the syngnathids enable the authors to disentangle impacts of sex (differentially costly gamete types) and costs of parental care upon immune function.

The study design is clever, comparing (for each of 3 species) immune and metabolic gene expression by sex and by differential investment in parental care (e.g., pregnant vs. non-pregnant males). To frame each comparison, the authors lay out clear hypotheses and associated rationales. The study also appears to have been carefully executed, though (as the authors note) it might have been even more interesting to compare whole transcriptomes rather than candidate genes.

Key findings were that there were no gene expression differences by sex and thus the predicted gradient in sexual dimorphism was not apparent. However, the authors do report a tradeoff between resource allocation and immune gene expression in pregnant males. These results are interesting indeed. The authors then provide a thoughtful discussion, including links to existing theory and suggestions for novel explanations for their findings (e.g. adaptation to provision of parental care potentially reducing its metabolic cost, lines 459-460). I also appreciated the extremely intriguing comparison of mammals and syngnathids re: cd45 in successful pregnancy (lines 474-478).

I just would suggest that the authors consider a few minor revisions, as follows:

1) Perhaps address implications of the fact that the animals were wild caught individuals with unknown immunological and reproductive histories. Might that have confounded their gene expression patterns? Might the power of a future study be enhanced if immune responses were experimentally induced with a controlled exposure of each of the animals?

2) Perhaps explain why the gills were chosen for RNA extraction?

3) The gigantic paragraph beginning line 463 meanders through a lot of different observations and interpretations and it’s easy to lose the thread. Perhaps streamline or reorganize to improve clarity?

4) The figures came through in inverse order and were very blurry so something was amiss in the formatting. Nonetheless, I felt that figure 4 was by far the strongest and most compelling. Perhaps figures 1-3 could be altered to make the presentation of information more efficient?

6. PLOS authors have the option to publish the peer review history of their article (what does this mean?). If published, this will include your full peer review and any attached files.

Reviewer #1: Yes: Nuno Monteiro

Reviewer #2: No

---

## [Author Response · Author response to Decision Letter 0]

23 Jun 2020

Point-by-point responses to reviewers comments

1. Is the manuscript technically sound, and do the data support the conclusions?

Reviewer #1: Yes

Reviewer #2: Yes

2. Has the statistical analysis been performed appropriately and rigorously?

Reviewer #1: Yes

Reviewer #2: Yes

3. Have the authors made all data underlying the findings in their manuscript fully available?

Reviewer #1: Yes

Reviewer #2: No

4. Is the manuscript presented in an intelligible fashion and written in standard English?

Reviewer #1: Yes

Reviewer #2: Yes

5. Review Comments to the Author

Reviewer #1:

I have read your manuscript with great interest, as I find the topic fascinating. The text is clear and easy to follow (at least to me), the methodology is appropriate, the figures are informative and the results/discussion very interesting. As such, I congratulate the authors on a really nice work. I list bellow some questions and very minor suggestions that might, if the authors agree, possible slightly improve the manuscript.

P3 L68: “Syngnathids, the teleost family of seahorses and pipefishes, …”

Unimportant but, to be more wide-ranging, would you consider including at least the seadragons?

Answer: We have added the seadragon to the list of Syngnathids

Line 68: “Syngnathids, the teleost family of seahorses, pipefishes and seadragons, …

P4 L95: “Nerophis ophidion (straight-nosed pipefish) with a comparably low paternal investment, as females simply attach their eggs on the abdomen of the male without any additional provisioning of paternal investment”.

If you are stating that females don’t invest further once the oocytes are passed onto the male, I agree. If you are stating that the males don’t provision during pregnancy, I have my doubts. Either way, maybe you could refine this sentence in order to avoid any hypothetical misperception.

Answer: We want to thank the Reviewer and have changed this statement.

Line 95: “Nerophis ophidion (straight-nosed pipefish) evolved towards a comparably low paternal investment, as females attach their eggs on the abdomen of the male without the development of a placenta-like structure in the male.”

P9 L208: “Statistical analysis of gene expression was done calculating two PERMANCOVA…”

Sounds weird… was done by? Or something like that.

Answer: We have now included a “by” and have changed the “two” to the appropriate “a”.

L211: “Statistical analysis of gene expression was done by calculating a PERMANCOVA …”

P10 L229: “… nor Syngnathus typhle show a sexual dimorphism in gene expression…”

Maybe remove the ‘a’ before sexual dimorphism…

Answer: We have removed the unnecessary “a” before sexual dimorphism.

L231: “As such, neither Nerophis ophidion, Syngnathus rostellatus nor Syngnathus typhle show sexual dimorphism in gene expression patterns of the measured immune system-related or metabolism-related genes.”

P17 L368: repeated ‘in the’

Answer: We deleted the first “in the” from the sentences.

L369: “Such genes can be found in the adaptive immune system gene group as this group shows differential expression depending on male gravity stages in both S. rostellatus and S. typhle.”

P20 L431: “Second, we hypothesized that the adaptation to intense parental care, measured as difference in sexual dimorphism between species, increases with the extent of adaptation to paternal care”.

Don’t really understand this sentence…

Answer: We have changed this sentence.

Line 432: “Second, we hypothesized that the ability to provide intense parental care, measured as the difference in sexual dimorphism between species, increases with the extent of adaptation to parental care. “

P21 L479: “In Syngnathus typhle, we can see effects of male gravity stage on…”

As you probably detected by now, I am not a native English speaker. Such, my ability to provide significant improvements to the text is very slim. Nevertheless, I was wondering if the term ´gravity´ is really a synonym of pregnancy… If it is, I am sorry to have wasted your time.

Answer: The term gravity was modified from the term gravid which is synonymous for pregnant. We used this term as it can be used to describe the stage of the cycle a female (or male in this case) undergoes changing from non-pregnant over the different pregnancy stages to after pregnant, this seemed ideal because it spans all three terms (non-pregnant, pregnant and after pregnancy) in a on word description.

P21 L460: “As such, species with less-specialised adaptation of life-history traits to parental care could eventually have a higher cost associated with the provisioning of parental care”.

I don’t disagree with this ‘speculation’ (as characterized by the authors), at all. Indeed, this would be something interesting to research further (there is not much information on syngnathid male investment during pregnancy). In a previous study (Pregnant pipefish with a simple brooding surface lose less weight when carrying heavier eggs: evidence of compensation for low oocyte quality?), I found that in a different Nerophis species (N. lumbriciformis), males seem to invest considerably during pregnancy, especially so when eggs are of poorer quality. I’m not asking for a citation (really!), just saying that your ‘speculation’ seems to be a nice hypothesis (backing up the observed change in the expression of metabolism-related genes group in Nerophis).

Answer: Your study is exactly in line with what we find in N. ophidion and we now refer to your study in our manuscript. It would be fascinating to investigate what triggers a male to invest more into a specific brood. Generally, it seems that simple brooding structure pipefish are completely understudied organism as they definitely seem to be suitable to answer more questions about sex role reversal, parental investment, sexual dimorphism and sexual selection.

Line 459: “This is supported by previously documented metabolic costs (high paternal weight loss during care for small weight brood) of parental care in Nerophis lumbriciformis (21).”

P22 L486: “As in N. ophidion this could hint towards a depletion of resources during pregnancy that hamper with a fast recovery of the male after pregnancy has ceased.”

I agree. Having worked with both Syngnathus and Nerophis, we see that Nerophis males that much more time to get pregnant again. For instance, Syngnathus abaster males can become pregnant rapidly after giving birth while N. lumbriciformis males need many days (in the lab, more than a week, at best). In the field, based on their migration patterns to and from the intertidal, they seem to take a month or more to become pregnant again.

Answer: Wow, those are long times scales to get pregnant again, especially considering the duration of the pregnancy and the confined length of the breeding season. We could, however, not find a publication where this pattern of pregnancy break is described.

P6 L150: “The above-mentioned pattern was expected to be strongest in S. typhle, intermediate in S. rostellatus and…”

Out of curiosity, what do you think is the main driver(s) of the differences observed between the two Syngnathus species?

Answer: S. rostellatus has a shorter pregnancy with lesser eggs and at birth, the juveniles are at an earlier developmental stage and smaller than in S. typhle. Supposedly, S. rostellatus males thus invest less in their offspring in contrast to S. typhle.

P22 L499: male instead of mal

Answer: The “e” was added.

L500: “This differential expression of adaptive immune system genes during intense male pregnancy could hint towards the evolutionary adaptations to paternal pregnancy.”

P7 L171: “At mid pregnancy, …”

Since pipefish (those with a marsupium) pregnancy can, roughly, be sub-divided into two periods (embryos within the egg and outside the egg but still inside the male’s brood pouch), and male investment can differ within these two stages (I imagine that male investment should be more critical once the embryos hatch, having spent most of the mother-derived resources), I would ask if 1) the embryos from both Syngnathus species were within the same stage of development? and 2) the embryos were still within the egg or already free in the pouch? Personally, I think it would be helpful to state this information in the methods. If I’m correct, and male investment is stronger once the embryos hatch (I think I discuss something along these line in the “woman in red” paper), sampling before and after embryo hatching would produce very different results.

Answer: It definitely makes sense, that male investment changes during the course of pregnancy, especially in species with a placenta-like system. May there even be three different parts of the pregnancy? The two mentioned already above and a third intermediate stage where the juveniles have already hatched but still rely on the yolk sack. We are currently investigating the developmental stages in detail for the species mentioned throughout this study. However, we are confident that for the Syngnathus species similar developmental stages were used. Due to the very distinct pregnancy in N. ophidion, we gave our best to use comparable developmental stages of the two Syngnathus species and the Nerophis species. 

Line 172: “At mid pregnancy, when Syngnathus embryos had hatched in the brood pouch, males were either left in the tanks to be sampled after parturition or sampled immediately”

P23 L513: “A prevention of embryo rejection as non-self-tissue or potential immune priming of the offspring might explain the identified pattern”.

I agree. This being said, what I’m about to ask adds nothing to your manuscript, but I imagined I have a chance to hear your opinion, in case you are willing to share your thoughts on the subject. Since fertilization occurs within the brood pouch, a sneaker male strategy (never found… probably for a reason) would be nearly impossible (a sneaker male would have a huge task, Tom Cruise ´mission impossible´ style, to be able to inject his few spermatozoa together with a female laying her oocytes. The time window would be extra-small). In Nerophis, although difficult, a sneaker male (I have been unable to find any instances of this as well) would probably have more time and easier access to unfertilized eggs. So, do you think that the immune system somehow mirrors this difference in the chance of rearing an embryo that is from a different father?

Answer: This is an interesting question. So far we have never doubted security of parentage. I would guess, that if parentage is not secured, investment in the brood would be lower, thus, in Nerophis more resources would be available after pregnancy to be invested into own metabolism or immune defense. Hence, if sneaker males existed, this might add to the explanation of why we could not find immune gene expression differences among the reproductive stages in the N. ophidion.

 According to our hypotheses, we expected that embryo rejection and immune priming would only occur in Syngnathus species due to the placenta-like tissue. In case there was direct provisioning of nutrient, oxygens from the father to the embryo Nerophis would also need to prevent non-self-rejection of the embryo and might also transfer immunological information to their offspring.

Again, compliments on your work.

Nuno Monteiro

Reviewer #2:

Does the manuscript adhere to the PLOS Data Policy? Additional details can be found at http://journals.plos.org/plosone/s/materials-and-software-sharing.

I don’t know

Answer: We will provide all Data generated and used for this study after acceptation by PLOS in the Earth science digital data library PANGAEA.

This manuscript unites a strong and fascinating conceptual foundation with the unique biological system of syngnathids – which exhibit male pregnancy across a spectrum of investment intensities – to investigate causes of sexual dimorphism in immune function. As the authors astutely note (lines 75-76), the syngnathids enable the authors to disentangle impacts of sex (differentially costly gamete types) and costs of parental care upon immune function.

The study design is clever, comparing (for each of 3 species) immune and metabolic gene expression by sex and by differential investment in parental care (e.g., pregnant vs. non-pregnant males). To frame each comparison, the authors lay out clear hypotheses and associated rationales. The study also appears to have been carefully executed, though (as the authors note) it might have been even more interesting to compare whole transcriptomes rather than candidate genes.

Answer: We agree, that a comparative transcriptomic approach would have given more detailed insights about gene expression changes among the different species over male pregnancy. Our group is currently working on a transcriptomic approach including an even higher number of species allowing to address male pregnancy evolution, parental investment and sexual dimorphism.

Key findings were that there were no gene expression differences by sex and thus the predicted gradient in sexual dimorphism was not apparent. However, the authors do report a tradeoff between resource allocation and immune gene expression in pregnant males. These results are interesting indeed. The authors then provide a thoughtful discussion, including links to existing theory and suggestions for novel explanations for their findings (e.g. adaptation to provision of parental care potentially reducing its metabolic cost, lines 459-460). I also appreciated the extremely intriguing comparison of mammals and syngnathids re: cd45 in successful pregnancy (lines 474-478).

Answer: We want to thank the Reviewer for the supportive words.

I just would suggest that the authors consider a few minor revisions, as follows:

1) Perhaps address implications of the fact that the animals were wild caught individuals with unknown immunological and reproductive histories. Might that have confounded their gene expression patterns? Might the power of a future study be enhanced if immune responses were experimentally induced with a controlled exposure of each of the animals?

Answer: We agree, that individual reproductive and immunological histories might have an impact on the gene expression measured in our fish. However, we are looking at the gene expression relative to the trait of interest (parental investment / sex), so as long as all tested individuals are from the wild and even a similar environment (as granted here) at least the immunological history would be similar among all individuals. Agreeably, noise in the gene expression data generated through natural variability by different individual histories might mask patterns of gene expression. Thus, one could argue, that sex related differences are masked by individual immunological histories. But the fact, that patterns of parental investment during paternal care, are still visible shows that the signal is strong and thus even strengthens their explanatory power.

Our goal was to look at the changes in not particularly immunologically induced individuals to see changes in the gene expression pattern inferred by sex and reproductive stages. However, studies measuring induced/ enhanced immune responses over multiple generations (and thus also known immunological histories) were done in S. typhle in our lab previous to this study (See Beemelmanns & Roth, 2016 “Bacteria-type-specific biparental immune priming in the pipefish Syngnathus typhle”, Evology and Evolution 6(18), 6735-6767 and Beemelmanns & Roth, 2017 “Grandparental immune priming in the pipefish Syngnathus typhle” BMC Evolutionary Biology 17:44).

2) Perhaps explain why the gills were chosen for RNA extraction?

Answer: Gill tissue was used as it is supposed to be a highly vascularized tissue and thus expression of immune genes is supposed to be close to the systemic expression of immune genes. Further, using gill tissue, renders our results comparable to other studies done within our group.

3) The gigantic paragraph beginning line 463 meanders through a lot of different observations and interpretations and it’s easy to lose the thread. Perhaps streamline or reorganize to improve clarity?

Answer: We have tried to streamline the mentioned paragraph without losing the information we think are of importance.

Line 471: “The gene expression pattern is affected by male gravity stage in S. rostellatus illustrated by a distinct gene expression in pregnant S. rostellatus compared to non-pregnant males and males after parturition. This, however, also suggests a fast recovery to a state similar to pre-pregnancy possibly attributed to the close contact with the juveniles. A modulation of the immune system might be necessary to prevent juvenile rejection but also to allow transfer of immune components from the male to its offspring. One of the three genes with high contribution in both syngnathid species, cd45, is associated with fetal rejection and subsequent miscarriage in human pregnancy upon upregulation (28). Thus, a downregulation of cd45 expression could be important in syngnathids for the tolerance of the embryos and the completion of a successful male pregnancy. “

4) The figures came through in inverse order and were very blurry so something was amiss in the formatting. Nonetheless, I felt that figure 4 was by far the strongest and most compelling. Perhaps figures 1-3 could be altered to make the presentation of information more efficient?

Answer: We are apologizing for the blurriness and will be more careful when we upload the revision. We agree, that Figure 4 is the most intuitive to interpret as single genes are shown in their up and down regulation. However, the depiction of gene up and down regulation is also misleading as it is in need for a reference. This reference could be a functional cascade (upregulation of gene A could be advantageous but upregulation of gene B could be deleterious for the same mechanism) or known expression of the gene from literature. Also, bar charts all genes by itself, this makes them easily understood but does not incorporate the complexity of factors working together in potentially opposite gene expression directions. Thus, showing all the graphs as bar charts would lead to 33 bar charts for each species, which then are a lot more difficult to interpret as effects might be missed or overrepresented.

The principal component analysis, used for Figure 1-3, has the advantage to entail an analytic tool and a graphic tool in one. As such we are able to see how the overall variance is spread in the system, how individual expression is distributed and with this also the connectivity (how they relate to each other) and which genes are adding what to the total variance and thus infer gene importance through gene contribution. This is why we have chosen the PCA as graphical representation of gene expression of each species.

6. PLOS authors have the option to publish the peer review history of their article. If published, this will include your full peer review and any attached files.

Do you want your identity to be public for this peer review? For information about this choice, including consent withdrawal, please see our Privacy Policy.

Reviewer #1: Yes: Nuno Monteiro

Reviewer #2: No

---

## [Decision Letter · Decision Letter 1]

21 Jul 2020

Parental investment and immune dynamics in sex-role reversed pipefishes

PONE-D-20-02201R1

Dear Dr. Tanger,

We’re pleased to inform you that your manuscript has been judged scientifically suitable for publication and will be formally accepted for publication once it meets all outstanding technical requirements.

Kind regards,

Cheryl S. Rosenfeld, DVM, PhD

Section Editor

PLOS ONE

Additional Editor Comments (optional):

Reviewers' comments:

Reviewer's Responses to Questions

**Comments to the Author**

1. If the authors have adequately addressed your comments raised in a previous round of review and you feel that this manuscript is now acceptable for publication, you may indicate that here to bypass the “Comments to the Author” section, enter your conflict of interest statement in the “Confidential to Editor” section, and submit your "Accept" recommendation.

Reviewer #1: All comments have been addressed

Reviewer #2: (No Response)

2. Is the manuscript technically sound, and do the data support the conclusions?

Reviewer #1: Yes

Reviewer #2: Yes

3. Has the statistical analysis been performed appropriately and rigorously? 

Reviewer #1: Yes

Reviewer #2: Yes

4. Have the authors made all data underlying the findings in their manuscript fully available?

Reviewer #1: Yes

Reviewer #2: Yes

5. Is the manuscript presented in an intelligible fashion and written in standard English?

Reviewer #1: Yes

Reviewer #2: Yes

6. Review Comments to the Author

Reviewer #1: Thanks for the responses and for sharing some of your thoughts on this fascinating subject. The manuscript seems ready for publication.

Reviewer #2: Thanks for your revision to the discussion, as I find it much clearer now. Thanks also for explaining the rationale for your choice of figures.

As for the rest, I do appreciate your explanations in response to my questions about immunological histories and the choice of gills. But please note that I was suggesting that you should explain those points in the revised manuscript, and as far as I can tell, you chose not to change the manuscript to address those concerns.

To make the article as compelling as possible to people who do not work on the same system you do, I still think it would be stronger with those explanations included. But it is up to you. This remains a strong paper.

7. PLOS authors have the option to publish the peer review history of their article (what does this mean?). If published, this will include your full peer review and any attached files.

Reviewer #1: **Yes: **Nuno Monteiro

Reviewer #2: No

---

## [Editor Report · Acceptance letter]

16 Sep 2020

PONE-D-20-02201R1

Parental investment and immune dynamics in sex-role reversed pipefishes

Dear Dr. Tanger:

I'm pleased to inform you that your manuscript has been deemed suitable for publication in PLOS ONE. Congratulations! Your manuscript is now with our production department.

Kind regards,

on behalf of

Dr. Cheryl S. Rosenfeld 

Section Editor

PLOS ONE